# Associations between Diet Quality and Anthropometric Measures in White Postmenopausal Women

**DOI:** 10.3390/nu13061947

**Published:** 2021-06-06

**Authors:** Andrea Y. Arikawa, Mindy S. Kurzer

**Affiliations:** 1Department of Nutrition and Dietetics, University of North Florida, Jacksonville, FL 32224, USA; 2Department of Food Science and Nutrition, University of Minnesota, Saint Paul, MN 55108, USA; mkurzer@umn.edu

**Keywords:** healthy eating index, calcium, waist circumference, diet history questionnaire, supplements, menopause, bone mineral density

## Abstract

The purpose of this cross-sectional study was to examine the relationship between diet and anthropometric measures in postmenopausal women. Data collected from 937 women enrolled in the Minnesota Green Tea Trial (NTC00917735) were used for this analysis. Dietary intake and health-related data were collected via questionnaires. Body weight, height, and waist circumference (WC) were measured by the study staff. The mean age of participants was 59.8 years and mean WC was 83 cm. Approximately 30% of the participants had WC greater than 88 cm. Healthy Eating Index-2015 score was 72.6 and the Dietary Inflammatory Index score was 0. Intakes of whole grains, dairy, protein, sodium, and saturated fat did not meet the dietary guidelines. Only 12.5% consumed the recommended daily amount of calcium (mean intake = 765 mg/day). When calcium supplements were considered, only 35.2% of the participants had adequate intakes, even though 68.9% reported taking a calcium supplement. We found that age and number of medications taken were significantly associated with waist circumference (*p* = 0.005). Women who reported taking two or more medications had greater WC (85 cm) compared to women who reported not taking any medications (82.2 cm), *p* = 0.002. Our findings suggest that achieving adequate calcium and vitamin D intake may be challenging to postmenopausal women.

## 1. Introduction

The transition to menopause is accompanied by several concomitant changes in women’s biological, physiological, behavioral, and social characteristics that will define women’s well-being and future risk of disease. One of the major physiological changes that occur with menopause is a marked decline in estradiol production, which is associated with several adverse consequences such as sleep disturbances, vasomotor symptoms such as hot flashes and night sweats, sexual dysfunction, metabolic disorders, bone loss, dyslipidemia, and obesity [1,2]. 

Approximately 75% of women ages 60 years and older are overweight [3]. Previous research indicates that fat mass increases during the years preceding the transition to menopause and abruptly worsens at the onset of menopause, followed by stabilization [4]. Although these observed increases in fat mass seem to be related to aging, rather than menopause, evidence suggests that central fat accumulation is higher in postmenopausal women compared with premenopausal women [3], which is attributed to the decreased levels of estradiol and consequent increases in the activity of androgen receptors in visceral adipose tissue [5]. In addition to changes in fat mass, a sharp menopause-related increase in low-density lipoprotein (LDL)-cholesterol has been reported in the cohort Study of Women’s Health Across the Nation (SWAN) [6], which could predispose postmenopausal women to cardiovascular disease and other chronic diseases such as diabetes. Indeed, menopause has been recognized as a female-specific risk factor for cardiovascular disease by the American Heart Association [7].

Bone loss is considered a hallmark of menopause. A decline in bone mineral density during the transition to menopause has been well documented in prospective cohort studies [8,9]. The role of diet on the prevention of bone mineral loss is not entirely elucidated in light of conflicting findings regarding the benefits of calcium and vitamin D supplementation for the prevention of fractures [10,11,12]. Nonetheless, adequate intake of these two nutrients is recommended for the prevention of bone loss and fractures in postmenopausal women [13]. Adequate protein intake also seems to be associated with bone mineral density. Protein intake representing around 24% of total energy intake has been associated with slower rates of bone loss and lower risk of hip fractures in older individuals diagnosed with osteoporosis [14]. Finally, regular physical activity has also been shown to positively affect bone health in women [15]. 

While some of the natural changes that occur with menopause are not modifiable, lifestyle behaviors, particularly dietary behavior and physical activity, can be modified to reduce the impact of disease risk factors on women’s health such as weight gain, dyslipidemia, and bone loss, among others. To target dietary behavior as a means of lowering the risk of disease in postmenopausal women, it is necessary to identify specific diet-related concerns in this population. Several measures of diet quality can be used to assess dietary patterns of postmenopausal women; the Healthy Eating Index (HEI) is a commonly used index of diet quality because it reflects dietary adequacy in relation to the Dietary Guidelines for Americans [16]; the Dietary Inflammatory Index (DII) assesses the inflammatory potential of the diet on a continuum from anti-inflammatory, represented by lower scores to pro-inflammatory, represented by higher scores [17]. These indices are useful in the context of large datasets because they provide a measure of overall dietary complexity, which can be analyzed in conjunction with intakes of individual nutrients to obtain a more thorough assessment of an individual’s diet. The purpose of this study was to examine relationships between diet quality as assessed by the HEI and the DII and anthropometric measures in a large sample of postmenopausal women living in a Midwestern state of the U.S.

## 2. Materials and Methods

Baseline data collected from 937 postmenopausal women enrolled in the Minnesota Green Tea Trial (MGTT) were used for this cross-sectional analysis. The MGTT was a double-blinded placebo-controlled trial that assessed the effects of green tea catechin supplements on biomarkers of breast cancer risk (clinical trial ID: NCT00917735) [18].

### 2.1. Study Participants

Eligible participants had to be between the ages of 50 and 70, report being generally healthy, postmenopausal for at least one year, not using any hormone therapy within the past six months, and having been diagnosed with heterogeneously or extremely dense breasts on a recent screening mammogram. Those who reported using hormone replacement therapy, elevated liver enzymes, or over than 5 kg change in body weight over the past year were not eligible. Written consent was obtained from all participants and the trial protocol was approved by the University of Minnesota Institutional Review Board. Details about participant selection and enrollment into the MGTT have been published previously [18].

### 2.2. Dietary Assessment

Dietary intake was assessed using the National Cancer Institute’s diet history questionnaire (DHQ-1) [19]. This questionnaire included 124 food items with details about portion size and frequency of consumption including vitamin and mineral supplements. 

Data originated from the DHQ-1 was used to calculate the Healthy Eating Index (HEI)-2015, according to the method previously described [20]. Briefly, the HEI is a measure of diet quality based on alignment of dietary components to the recommendations of the Dietary Guidelines for Americans [16]. The HEI includes 13 components from which the sum can result in scores ranging from 0 to 100. The higher the HEI score, the greater the alignment between the individuals’ diets and dietary recommendations. Previous research has shown that the HEI is valid and reliable when tested using dietary data from a representative sample of Americans [21].

DHQ-1 data were also used to calculate the Dietary Inflammatory Index (DII) score, as described by Shivappa et al. [17]. The *z*-scores for intakes of 28 food parameters were calculated (calories (kcal), carbohydrate (g), protein (g), dietary fiber (g), saturated fat (g), trans fats (g), monounsaturated fat (g), omega-6 (g), omega-3 (g), cholesterol (mg), alcohol (g), vitamin A (µg), β-carotene (µg), thiamin (mg), niacin (mg), riboflavin (mg), vitamin B6 (mg), folate (µg), vitamin B12 (µg), vitamin C (mg), vitamin D (µg), vitamin E (mg), magnesium (mg), iron (mg), zinc (mg), selenium (mg), caffeine (mg), isoflavones (mg)). Percentile scores were calculated from the *z*-scores and then doubled and ‘one’ was subtracted from the values to achieve a symmetrical distribution. Next, each food parameter’s centered percentile value was multiplied by an ‘inflammatory effect score’ and all scores for the 28 food parameters were summed to obtain the overall DII for each study participant. Intake of nutrients via supplements was not included in the calculation of the DII. Although the original DII includes 45 food parameters, several of these were not available from the DHQ-1. However, it can be argued that food components that were not included in the calculation of the DII are consumed in low amounts in Western diets such as ginger, saffron, turmeric, rosemary, flavonols, etc., and therefore, the absence of these components will likely have a negligible effect on the overall DII scores of the present population. In addition, previous published studies have also utilized this approach when calculating the DII [22,23]. Higher DII scores indicate the higher inflammatory potential of the diet.

### 2.3. Anthropometric Measures

Body weight was measured to the nearest 0.1 kg using a digital scale and height was measured to the nearest 0.1 cm with a wall-mounted stadiometer. Waist circumference was measured at the uppermost lateral border of the iliac crest at the narrowest point of the torso, using a flexible body tape. All measurements were done by trained research staff.

### 2.4. Bone Mineral Density

Bone mineral density (BMD) in g/cm^3^ was measured by dual x-ray absorptiometry (DXA) in a subsample of 121 study participants. Whole body DXA scans were performed using a GE Healthcare Lunar iDXA (GE Healthcare) and analyzed with the Encore software v. 13.6, revision 2. *T*-scores were obtained by using peak bone mass from the manufacturer’s reference population (age 30). *T*-scores less than −1.0 and −2.5 indicate osteopenia and osteoporosis, respectively, according to WHO criteria [24].

### 2.5. Other Questionnaires

A comprehensive health questionnaire was completed by study participants about demographics, physical activity (how many days per week did participant exercise for at least 20 min), lifestyle factors, reproductive health history, medication, and supplement use. 

### 2.6. Statistical Analyses

Descriptive statistics were generated by calculating frequencies and percentages for categorical variables and means and standard deviations for continuous variables. We examined relationships between the continuous variables of interest using Pearson’s correlations. Physical activity was reported in days per week of exercise and for data analysis, a dichotomous variable was created as follows: high physical activity (at least four days per week of 20 min of physical activity) and low physical activity (less than four days per week of 20 min of physical activity).

A multiple linear regression model using the backward method was fit to identify which variables were most related to WC, and the initial model included age (years), medication use category, as described below, parity (yes/no), exercise (low/high), HEI score, DII score, years since menopause, fat intake (g), and glycemic load of the diet. All variables were checked for multicollinearity prior to inclusion in the initial model. Post-hoc power analysis for the multiple regression model was calculated and the observed statistical power was 0.999.

To further explore the relationship between WC and medication use, analysis of covariance was performed including age as a covariate. Medication use was classified into three categories: low (if participants reported taking one medication or none, regularly), medium (if participants reported taking 2 to 4 medications, regularly), and high (if participants reported taking five or more medications, regularly).

To examine the relationship between diet quality and WC, linear models were fit adjusting for age and using HEI score quartile as the independent variable. We also used Pearson’s or Spearman correlations and linear models to explore relationships between BMD, dietary variables, and physical activity in a subsample of participants who completed DXA scans. All *p*-values < 0.05 were considered statistically significant and adjustment for multiple comparisons was done by the Bonferroni test. All analyses were performed using IBM SPSS version 26.0 (IBM Corp., Armonk, NY, USA).

## 3. Results

### 3.1. Study Participants

Table 1 shows the selected characteristics of the study population. Dietary data were available for all 937 participants, but information about years since menopause and physical activity was missing for 43 and 123 participants, respectively. The majority of participants were White (97.2%) and less than 1% were Hispanic. Mean BMI was 25.1 kg/m^2^ and 55.6% of the participants were classified as normal weight, while 33.5% and 10.9% were classified as overweight and obese, respectively. Between 7 and 27% of participants took a prescription medication from the categories listed in Table 1. Noticeably, while 34.7% did not report taking any prescription medications, 28.5%, 18%, and 19.5% reported taking one, two, or three or more prescription medications, respectively. 

### 3.2. Dietary Intake

The mean HEI score for the participants was 72.6 (Table 1). Figure 1 depicts a radar plot of the thirteen components used to calculate the HEI stratified by WC category. The plot indicates that although the consumption of fruits, vegetables, seafood, and plant proteins, refined grains, and added sugars were within 90% of the recommended amounts, consumption of whole grains, dairy, total protein, sodium, and fats were inadequate. Consumption of whole grains and sodium met only 40% of the recommended amounts, while consumption of total protein met 70% of recommendations. In addition, the mean daily dietary calcium intake was 764.5 mg, indicating that this population was consuming only 63.7% of the recommended 1200 mg per day for women older than 51 years [16]. The intake increased to 1048 mg per day, when calcium supplements were added, which was still not sufficient to reach the dietary recommendation. Only 12.5% of participants had adequate intakes of calcium without counting with supplements. When calcium supplements were considered, 35.2% of the participants had adequate intakes, even though 68.9% reported taking a calcium supplement. Mean intakes of vitamin D from dietary sources and supplements were 3.33 µg and 3.56 µg per day, respectively. More than half of the participants reported taking a vitamin D supplement (66.6%). Mean total vitamin D was 6.88 µg, indicating that less than 50% of the recommended amount was met. 

HEI score was inversely associated with DII (r = −0.347, *p* < 0.001) and waist circumference (r = −0.152, *p* < 0.001), while DII was strongly and negatively associated with glycemic load (r = −0.676, *p* < 0.001). DII scores ranged between −4.54 and 4.81, with a mean DII score of 0. When we looked at DII scores and glycemic load stratified by quartiles of HEI score, we found a significant association between quartiles of HEI score and DII scores, F (3932) = 35.7, *p* < 0.001, η^2^ = 0.104. The findings indicated that mean DII scores were significantly lower for quartiles 2, 3, and 4 of HEI score, compared with quartile 1. No significant relationships were found between HEI scores and glycemic load (Table 2).

### 3.3. Waist Circumference

WC measurements were available for 935 of the 937 participants. The mean of WC was 83.6 cm. When classifying participants into high and low risk of chronic disease using a WC cut-off of 88 cm [25], it was found that approximately 30% of study participants were at high risk. There was also a significant relationship between WC and HEI quartiles; after adjusting for age, F (3925) = 10.7, *p* < 0.001. WC was lower as quartiles of HEI increased, as seen in Table 2. Figure 1 shows that the components of the HEI were very similar for women classified as high risk versus those classified as low risk of chronic disease as per WC values.

We examined whether age, medication use, physical activity category, dietary (HEI score, DII score, fat intake, glycemic load), and reproductive variables (years since menopause, parity) were associated with WC by fitting a multiple linear regression model. Overall, the variables selected were not found to be good predictors of WC, as evidenced by a small R^2^ = 0.054 (*p* < 0.0001) (Table A1). Noteworthy was the fact that mean WC for women who reported not taking any medications regularly was 82.2 cm (SE = 0.6) compared with 85.1 cm (SE = 0.6) for those who reported taking five or more medications regularly (*p* = 0.001). Women who reported exercising less than four times per week had higher mean WC (84.4 cm, SE = 0.5) compared with those who reported exercising four or more times per week (81.8 cm, SE = 0.4).

### 3.4. Bone Mineral Density

Mean BMD was 1.16 g/cm^3^. *T*-scores ranged from −1.4 to 3.4, but only two participants were classified as osteopenic, as indicated by *T*-scores below −1.0. There were no significant correlations between BMD and HEI, total calcium, or total vitamin D intake. BMD was significantly correlated with DII (r = −0.182, *p* = 0.046) and age (r = −0.223, *p* = 0.014), and there was a trend toward a significant correlation with physical activity expressed in days per week of at least 20 min of physical activity (r_s_ = 0.136, *p* = 0.052). 

## 4. Discussion

The present study aimed at investigating the relationships between diet quality, assessed by calculating the HEI-2015 and the DII, and other lifestyle variables collected from a large sample of postmenopausal women who participated in the MGTT [18]. Our population’s HEI score indicates that only 72.6% of the dietary guidelines were being met, with more pronounced deficits in whole grain, dairy, and total protein. Participants met recommendations for the intake of fruits and vegetables while total protein intake recommendations were not achieved, which is not in agreement with national data trends. Previously published data collected from the National Health and Nutrition Examination Survey (NHANES) in 2015–2016 showed that adults between 18 and 64 years had a HEI-2015 score of 58, with adequate intakes of total protein but noticeable deficits in fruit, vegetable, dairy, fatty acids, and whole grain intake [26]. Dairy intake in this study was 30% below the recommended amounts, which may explain the lower intake of calcium. Use of calcium supplements was reported by 68.9% of the participants and total calcium intake was improved significantly with the use of calcium supplements, increasing from 764 mg to 1048 mg (87.3% adequacy). When we looked at vitamin D intake, we found that only 45.9% of the recommended intake (15 µg/day) was met by study participants. It is thought that adequate calcium and vitamin D intake is especially important after menopause due to the decreases in bone mineral density associated with lower circulating estrogen [27], but controversy exists about the relationship between calcium and vitamin D intake and bone mineral density or risk of fractures. Bristow et al. [10] did not find any relationships between calcium intake and BMD in a cohort of osteopenic postmenopausal women over the age of 65 years. Similarly, a meta-analysis of randomized clinical trials indicated that the use of calcium and/or vitamin D supplements was not associated with a significant difference in risk of hip fractures compared to the placebo [28]. Findings from the Women’s Health Initiative (WHI) randomized controlled trial of calcium plus vitamin D supplementation indicated a 1% increase in hip bone density in the group that received 1000 mg calcium with 400 IU vitamin D, but no significant effects on hip fractures were found in the intent-to-treat analysis [29]. However, in subgroup analyses, it was reported that adherent women had a 29% reduction in hip fractures. Interestingly, this trial also found that supplements of calcium plus vitamin D increased the risk of renal calculi by 17%. Jackson et al. [30] also found that hip BMD was strengthened in postmenopausal women with the combination of calcium and vitamin D supplements. We also looked at the potential relationship between calcium and vitamin D intake and BMD data that was available for 121 participants in the present dataset, but we did not find any significant associations between BMD and calcium or vitamin D intake. These findings are not entirely surprising considering that vitamin D status is influenced by several factors including sun exposure, genetics [31], and obesity [32]. Even though vitamin D intake by study participants did not meet recommendations, vitamin D status may have been adequate, which would also support the normal BMD data described here. Notwithstanding, it is recommended that postmenopausal women, particularly those living in areas where sun exposure during winter months is not sufficient to produce vitamin D in the skin, consume adequate amounts of vitamin D through the diet and/or supplements when sufficient intake cannot be achieved. 

Physical activity was crudely measured in this study by asking study participants to report the number of days in which they engaged in at least 20 min of physical activity. It was found that approximately 45% of the women reported engaging in physical activity less than four times per week, while 55% engaged in four or more days per week of physical activity. In the subsample of 121 participants for whom a bone density scan was obtained, we found a nearly significant weak correlation between BMD and number of days per week of physical activity. These findings are in line with a large number of published studies examining the role of physical activity on BMD in postmenopausal women [15,33]. Unfortunately, we were not able to further explore the relationship between BMD and physical activity in the present report due to the small number of women for whom BMD data were available, and because physical activity was not assessed through a validated questionnaire or through more objective measures such as using a pedometer or an accelerometer.

DII score was calculated as another measure of diet quality in the present study. The mean DII score for our population was 0. This finding is in agreement with a previous study that reported a DII score of −0.62 for a subsample of postmenopausal women who participated in the WHI study [34]. In the WHI, DII score was significantly associated with levels of IL-6 and TNF-α-R2, which provided evidence for the validity of the DII score. Other studies have found significant associations between DII score and risk of disease such as colorectal cancer [22], cardiometabolic disease [35], and osteoporotic fractures [36]. Orchard et al. [36] used longitudinal data from the WHI to investigate the relationships between DII and risk of fractures as well as BMD measured by DXA. It was found that a higher DII score was associated with hip fracture risk only in White women younger than 63 years and the authors speculated that the much greater risk of fracture that occurs with older age would outweigh the benefits of a less inflammatory diet. The findings also indicated that lower loss of hip BMD occurred in women who consumed a less inflammatory diet (lower DII scores) over a six-year period [36]. Our findings seem to support this association between DII scores and BMD as we found a significant but weak negative correlation between BMD and DII (*r* = −0.182) in a subsample of our population. We also found a significant inverse correlation between DII score and HEI score. This was an expected finding, considering that higher DII scores indicate a more inflammatory diet, which would translate into a lower HEI. DII score was also strongly associated with glycemic load (*r* = −0.676, *p* < 0.001). To our knowledge, no previous studies have examined the associations between glycemic load and DII in postmenopausal women. In contrast with our study, Kim et al. [37] found that DII score was not correlated with glycemic load in a sample of 110 college students living in the southern U.S. These findings may be partially explained by the differences in dietary patterns between college students and postmenopausal women. For example, Kim et al. [37] reported HEI scores for female college students ranging between 34 and 61, while our HEI scores ranged from 42 to 92, indicating that postmenopausal women consume a higher quality diet.

WC and BMI were the primary anthropometric measures used in this study. Our data show that only 10.9% of study participants were classified as obese. This figure is almost four times lower than the prevalence data available for a representative sample of U.S. women, which indicated that 39.8% of non-Hispanic white women older than 20 were obese in 2017–2018 [38]. WC data indicated that 30% of participants were at high risk for disease using the cut-off value of 88 cm. Interestingly, HEI score as well as the components of the HEI were very similar between the high risk and the low risk women, indicating that differences in WC in this population may not be related to diet quality. One possible explanation for our findings could be that women who were interested in participating in the Minnesota Green Tea Trial were more likely to practice healthier lifestyle behaviors, which would lead to lower body weight. We also found that women who reported taking five or more medications regularly had higher WC compared with women who reported taking one or none, suggesting that higher WC was associated with the presence of risk factors for disease. Some of the most common medications reported by participants were anticoagulants, statins, antidepressants, blood pressure, and thyroid medications.

One limitation of the present study includes the racial background of study participants, which was primarily White and thus limits generalizability of our findings to other racial groups. It should also be noted that all women enrolled in the MGTT had been recently diagnosed with dense breasts, based on mammographic density. It is possible that these women showed higher HEI scores and lower DII scores than other populations because they were more concerned about breast cancer risk. Another limitation was the inclusion of only 28 out of the 45 food parameters used to calculate the DII score, which may have contributed to a less accurate estimation of the participants’ dietary inflammatory potential. However, as described earlier, we believe that the omission of certain foods might not have significantly affected the DII scores, considering the low intake of these foods in Western cultures. The questionnaire used to assess physical activity in this population was not a validated instrument, and as such, may not have accurately assessed physical activity in this population. Finally, data on BMD was only obtained for a small subsample of 121 women, which limited the power of statistical tests that were conducted using this variable.

## 5. Conclusions

Our findings support the notion that White postmenopausal women living in a U.S. Midwestern state and recently diagnosed with dense breasts engage in a healthier lifestyle compared to a national sample of women older than 20 years, as evidenced by higher HEI scores and lower prevalence of obesity. In contrast, it was found that intake of dairy, calcium, and vitamin D was not adequate in this population. BMD data did not indicate the prevalence of osteopenia or osteoporosis in a subsample of the participants and, despite the lack of evidence for a strong association between calcium intake and BMD [39], future studies are needed to shed light on the relationship between calcium and vitamin D intake and the health outcomes of postmenopausal women. 

## Figures and Tables

**Figure 1 nutrients-13-01947-f001:**
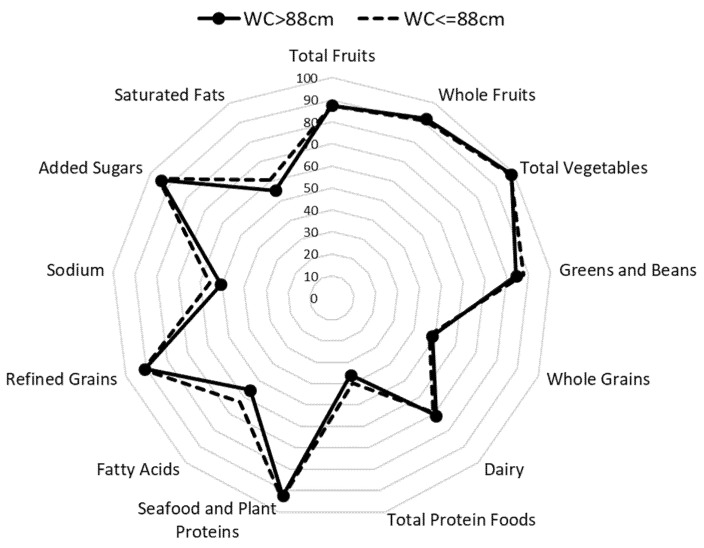
Radar plot showing the thirteen components of the Healthy Eating Index score stratified by WC for postmenopausal women who participated in the Minnesota Green Tea Trial. Mean HEI scores for those with WC > 88 cm and WC ≤ 88 cm were 73.7 and 75.4, respectively.

**Table 1 nutrients-13-01947-t001:** Selected characteristics of postmenopausal women who participated in the Minnesota Green Tea Trial.

Characteristic	*N*	Mean (SD) or Frequency (%)	(Min–Max)
Age (years)	937	59.8 (5.0)	(50.1–71.2)
Years since menopause	894	10.7 (7.4)	(1.0–40.1)
BMI (kg/m^2^)	932	25.1 (3.7)	(18.2–43.7)
Waist circumference (cm)	935	83.6 (10.2)	(60.3–120.5)
Healthy Eating Index (HEI)	937	72.6 (8.2)	(41.8–92.5)
Dietary Inflammatory Index (DII)	937	0.002 (2.2)	(−4.5–4.8)
Glycemic load	937	83.5 (32.8)	(16.4–256.9)
Protein (g)	937	58.2 (23.2)	(7.7–181.0)
Calcium from diet (mg)	937	764.5 (352.4)	(87.1–1950.3)
Calcium from supplements (mg)	937	283.2 (285.6)	(0–714.3)
Vitamin D from diet (µg)	937	3.33 (2.31)	(0.35–14.12)
Vitamin D from supplements (µg)	937	3.56 (3.32)	(0–7.14)
Physical activity	814		
Less than 4 days/week		363 (44.6)	
4 or more days/week		451 (55.4)	
Education	937		
Less than college degree		238 (25.4)	
College degree		420 (44.8)	
Graduate/Professional degree		273 (29.8)	
Parity	931		
No		218 (23.4)	
Yes		713 (76.6)	
Dietary Supplement Use	937		
No		115 (12.3)	
Yes		822 (87.7)	
Medications			
Depression/Anxiety	934	156 (16.7)	
Blood Pressure	935	187 (20.0)	
Statins	935	194 (20.7)	
Anticoagulants	935	253 (27.0)	
Thyroid	935	162 (17.3)	
Osteoporosis	935	76 (8.1)	

**Table 2 nutrients-13-01947-t002:** Dietary Inflammatory Index (DII), glycemic load, and waist circumference means stratified by quartiles of HEI score (*N* = 937).

Variables	HEI Quartiles			
	Q1 (41.8–67.4)	Q2 (67.5–73.2)	Q3 (73.3–78.3)	Q4 (78.4–92.5)
DII	0.90 (0.13)	0.36 (0.13)	−0.26 (0.13)	−0.99 (0.13)
*p*-values ^1^		0.028	0.000	0.000
Glycemic Load	86.14 (2.15)	80.63 (2.14)	80.84 (2.14)	86.27 (2.15)
*p*-values		NS	NS	
WC	85.89 (0.66)	84.49 (0.66)	83.39 (0.66)	80.73 (0.66)
*p*-values		NS	0.043	0.000

Abbreviations: DII: Dietary Inflammatory Index; HEI: Healthy Eating Index; NS: Non-Significant; WC: Waist Circumference. ^1^ *p*-values for comparisons with Q1.

## Data Availability

The data presented in this study are available on request from the corresponding author. The data are not publicly available because manuscripts are still being prepared by the authors.

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
