# Peer review of "Associations between Diet Quality and Anthropometric Measures in White Postmenopausal Women"

_nutrients, 2021, doi:10.3390/nu13061947_

Round 1

Reviewer 1 Report

The article concerns the relationship between the quality of diet and anthropometric variables in the group of postmenopausal women.

 Authors may consider the following comments:

The article is very interesting. short factual, mostly relating to the most important obtained results. The study tackles  two main  anthropometric  variables as  WC and BMD The second is  generally insignificant factor, so I would suggest to expose   more WC. The radar plot  is full of interest. However, I think it would be interesting to present this graph in two variants - for people with WC above and below the cut-off point. The same can be done in Table I wit concomitant presentation of the statistical differences. I think some dependencies would  be explained by such a this presentation of the data.  the data  needs to  be verified   for meeting  the condition of homogeneity of variance and normal distribution. If those criteria are not met , the data should be presented as median and lower-upper quartile.

Thank you.

Author Response

As per the reviewer’s suggestion, the radar plot was stratified by categories of WC. Given that both plots were very similar, we decided to keep table 1 as originally presented. Both distributions were checked for normality and homogeneity of variances and these assumptions were met. We modified the results section to incorporate the new radar plot findings (page 5, lines 183-184 and age 6, lines 215-216) and also added a sentence to the discussion (page 9, lines 333-335).

Reviewer 2 Report

Title: Associations between diet quality and anthropometric 2 measures in white postmenopausal women

This is a cross-sectional study with a large sample size. The aim this study is to examine the relationship between diet, anthropometric measures in postmenopausal women.

Methodology:

Further details of the methodology should be included to allow reproducibility of the study by other investigations. The authors should include information on the method of sample selection, the sample size calculation, the procedure for accessing the sample, and specify the periods of recruitment.

Results:

Report the number of persons in each phase of the study, i.e., the number of persons potentially eligible, screened for eligibility, confirmed as eligible, included in the study, completing follow-up, and analyzed. Please the authors should add a flow chart.

The authors say that there is no correlation between BMD and HEI, total calcium and total vitamin D intake. But is there a correlation between BMD and physical activity? I think the authors explain little about the influence of physical activity on BMD and waist circumference in the discussion. Please authors should add this both in the results section and in the discussion section.

Table 1: please put in the minimum and maximum interval between parenthesis and with hyphen, example (50.1-71.2).. In the title of table 1, the authors should put the meaning of MGGT

In the title of figure 1, the authors should put the meaning of MGGT

Author Response

Title: Associations between diet quality and anthropometric 2 measures in white postmenopausal women

This is a cross-sectional study with a large sample size. The aim this study is to examine the relationship between diet, anthropometric measures in postmenopausal women.

Methodology:

Further details of the methodology should be included to allow reproducibility of the study by other investigations. The authors should include information on the method of sample selection, the sample size calculation, the procedure for accessing the sample, and specify the periods of recruitment.

We decided not to include data related to sample selection and sample size calculation, because these data have been published as part of the design of the original trial. Since the data presented in the current manuscript are not part of the original trial’s design, we did not think it was appropriate to include these details in this manuscript. We added a sentence to section 2.1. of the methods, indicating that “details about the participant selection and enrollment into the MGTT have been published previously”. (page 2, lines 90-91)

Results:

Report the number of persons in each phase of the study, i.e., the number of persons potentially eligible, screened for eligibility, confirmed as eligible, included in the study, completing follow-up, and analyzed. Please the authors should add a flow chart.

We thank the reviewer for this suggestion. As described above, we included a sentence about participant enrollment under the section 2.1. of the methods (page 2, lines 90-91). We did not feel it was necessary to include these data in this manuscript, given that the data have already been published in a previous manuscript (reference #18).

The authors say that there is no correlation between BMD and HEI, total calcium and total vitamin D intake. But is there a correlation between BMD and physical activity? I think the authors explain little about the influence of physical activity on BMD and waist circumference in the discussion. Please authors should add this both in the results section and in the discussion section.

One major reason why we did not further explore BMD and physical activity in this manuscript was because BMD data was only available for a small subsample of the women (n=121). However, as per reviewer’s suggestion, we ran a Spearman correlation between number of days of physical activity and BMD in the sample of 121 women and the results are reported on page 6, lines 239-240. We also added a paragraph to the discussion, page 8, lines 288-300 to address these results.

The added text is highlighted in yellow.

Table 1: please put in the minimum and maximum interval between parenthesis and with hyphen, example (50.1-71.2).. In the title of table 1, the authors should put the meaning of MGGT.

We reformatted the min and max values and spelled out the acronym MGTT as per reviewer’s comments.

In the title of figure 1, the authors should put the meaning of MGGT

MGTT was spelled out in figure 1.

Round 2

Reviewer 2 Report

The authors have not responded to questions on methodology. 
The size of the sample calculation has to be done. The authors state that they have the calculation done according to a clinical trial they have published. This would not be adequate; the objective of this article is to examine relationships between diet quality as assessed by the HEI and the DII and anthropometric measures in a large sample of postmenopausal women living in a Midwestern state of the U.S. Therefore, the sample calculation should be done according to the objective of this manuscript. 
The details of the sample should be specified even if they have already been described in another articlle.

Author Response

The purpose of this article was to perform a secondary analysis of data collected from a large clinical trial. It is common practice in such analyses, to include all data available, therefore there is no point in calculating sample size. However, conducting a post-hoc power analysis gives us an idea of whether the large sample size of this study allowed us to have sufficient power to detect meaningful relationships among variables. As per the reviewer request, we added a sentence to the statistical analysis section indicating that the post-hoc power was observed to be 0.999 (page 4, lines 55-56). We hope this satisfies the reviewer's comment.

Thank you.